# On the Use of Wearable Face and Neck Cooling Fans to Improve Occupant Thermal Comfort in Warm Indoor Environments

Bin Yang [1,2], Tze-Huan Lei [3], Pengfei Yang [2], Kaixuan Liu [4] and Faming Wang [5,*]

1   School of Energy and Safety Engineering, Tianjin Chengjian University, Tianjin 300384, China; yangbin@xauat.edu.cn
2   School of Building Services Science and Engineering, Xi'an University of Architecture and Technology, Xi'an 710055, China; ypf@xauat.edu.cn
3   College of Physical Education, Hubei Normal University, Huangshi 435002, China; tzehuanlei@gmail.com
4   Apparel and Art Design College, Xi'an Polytechnic University, Xi'an 710048, China; liukaixuan819@163.com
5   School of Energy and Environment, Southeast University, Nanjing 211189, China
*   Correspondence: dr.famingwang@gmail.com

**Abstract:** Face and neck cooling has been found effective in improving thermal comfort during exercise in the heat despite the fact that the surface area of human face and neck regions accounts for only 5.5% of the entire body. Presently very little documented research has been conducted to investigate cooling the face and neck only to improve indoor thermal comfort. In this study, two highly energy efficient wearable face and neck cooling fans were used to improve occupant thermal comfort in two warm indoor conditions (30 and 32 °C). Local skin temperatures and perceptual responses while using the two wearable cooling fans were examined and compared. Results showed that both cooling fans could significantly reduce local skin temperatures at the forehead, face and neck regions by up to 2.1 °C. Local thermal sensation votes at the face and neck were decreased by 0.82–1.21 scale unit at the two studied temperatures. Overall TSVs decreased by 1.03–1.14 and 1.34–1.66 scale units at 30 and 32 °C temperatures, respectively. Both cooling fans could raise the acceptable HVAC temperature setpoint to 32.0 °C, resulting in a 45.7% energy saving over the baseline HVAC setpoint of 24.5 °C. Furthermore, occupants are advised to use the free-control cooling mode when using those two types of wearable cooling fans to improve thermal comfort. Finally, despite some issues on dry eyes and dry lips associated with those wearable cooling fans, it is concluded that those two highly energy-efficient wearable cooling fans could greatly improve thermal comfort and save HVAC energy.

**Keywords:** face cooling; neck cooling; personal thermal management; energy performance; perceptual responses; dry eye symptom





## 1. Introduction

Personal thermal management (PTM) has received tremendous attention in recent years because it helps save building energy and improve individual occupant thermal comfort [1–3]. In general, a personal thermal management system (PTMS) creates an ideal near-body thermal envelop so that individuals' thermal comfort could be improved. Furthermore, personal thermal management systems consume very little energy when compared to traditional HVAC (heating, ventilation, and air-conditioning) systems [4].

Wearable PTMS and non-wearable PTMS are the two types of PTMS. Non-wearable PTMS may include task ambient conditioning systems, personal comfort devices, and personalized ventilation systems [5–9]. Over the last 3 decades, there has been extensive research into the use of non-wearable PTMS to provide individual occupants thermal comfort [5–9]. Documented studies on non-wearable PTMS have clearly demonstrated that the use of non-wearable PTMS can improve individual thermal comfort in both non-air-conditioned and air-conditioned indoor environments [3]. Nonetheless, non-wearable

PTMS such as desk fans, ceiling fans and personalized ventilation systems are importable and inappropriate for active indoor occupants. In order to improve thermal comfort of active occupants, portable wearable PTMSs are needed. Individual thermal comfort could be improved, according to Yang et al. [3], if the intensified conditioning of personal micro-environment is moved closer to the human body. Hence, it is expected that wearable PTMS will improve individual thermal comfort while consuming little or no energy. Wearable PTMSs are currently divided into two categories: PTMS with cooling/heating modules and clothing made of specially designed materials and/or with a unique fabric layer structure. Ke and Wang et al. [10] investigated the effectiveness of nanoporous polyethylene (nanoPE) passive cooling clothing in improving occupants' indoor thermal comfort. It was found that the nanoPE passive cooling clothing could raise indoor acceptable air-conditioning setpoint temperature to 27.0 °C, saving 9–15% cooling energy. Ma et al. [11] conducted a numerical analysis of the energy saving performance of novel radiative cooling PTMS. According to the findings, personal radiative cooling textiles (with an air gap size of 5 mm) could save 4.6–12.8% energy in various cities around the world. Currently, only a few studies have reported the use of wearable PTMS combined with cooling units to improve occupant thermal comfort both locally and at the whole-body level. Song et al. [12] explored the effect of hybrid personal cooling clothing on thermal comfort of office workers in a hot indoor environment (34 °C, 65%RH [relative humidity]). The findings showed that hybrid cooling clothing could significantly improve both local- and whole-body thermal comfort. Udayraj et al. [13] evaluated and compared the performance of a traditional desk fan and air ventilation clothing with micro fans in three air temperatures (28, 30 and 32 °C). At all three air temperatures, the results showed that both systems had similar perceptual responses and skin temperatures. When compared to a traditional desk fan, air ventilation clothing could save 7–8% of energy. Wang et al. [14] investigated comfort of a thermally dynamic wearable thermoelectric wrist band. This device has the potential to improve overall thermal sensation, comfort and pleasantness by 0.5–1 scale unit. However, because the above results were obtained in thermal neutral conditions (<26.0 °C), the findings may not be applicable to warmer indoor conditions. On the other hand, air ventilation clothing had some practical limitations as reported in the aforementioned studies. For instance, such clothing became quite bulky during operation and some hygienic issues such as contaminated air due to sweating/body odor may not be avoided. As a result, there is a need to seek out better wearable personal cooling systems to improve local body cooling while working in indoor environments.

When using wearable PTMS for improving thermal comfort, local body thermal sensitivity to heat stress environments should be taken into account. According to Arens et al. [15], the head is insensitive to cold environments but sensitive to heat. Literatures [5,16] showed that face cooling can improve occupant thermal acceptability, shifting the acceptable upper boundary of indoor temperature. Cotter and Taylor [17] discovered that the head/face and neck regions have greater alliesthesial responses than the rest of the body. Despite the fact that the cooling area of face and neck regions is relatively small, a remarkable effectiveness on the thermal comfort improvement of human subjects could be expected due to high sensitivity at the face and neck areas. Thus, it was hypothesized that cooling of the head/face and neck region with wearable PTMS could significantly improve occupant thermal comfort while performing office work in warm indoor conditions.

It is also worth noting that current PTMS literatures frequently uses the fixed-power cooling module while failing to investigate the role of individual behavior response on PTMS [10,13]. It is well known that each occupant has a different preferred cooling temperatures. Given this, the individual free-control cooling module may be more effective than the fixed-power cooling method in improving thermal comfort improvement for indoor occupants while working in warm/hot environments. Contrary to the above expectation, Contrary to popular belief, Boerstra et al. [18] discovered that task performance was improved when participants had no control over personal desk fans as opposed to

free control. Hence, additional research is needed to address and compare the effects of free-control and fixed-power (i.e., no control) on occupant thermal comfort.

In this study, two types of highly energy efficient (power consumption $\leq 4$ W) wearable cooling fans (face cooling fan and neck cooling fan) were chosen to investigate their actual performance on the enhancement of occupant thermal comfort while performing office work in two warm indoor conditions. The effects of these two wearable cooling systems on overall and local physiological and perceptual responses of occupants were thoroughly investigated. In addition, the impact of face and neck cooling on thermal comfort was compared and discussed. Finally, two cooling control modes, fixed-power and free-control modes, were chosen to investigate how did personal control mode affected occupant local skin temperatures and perceptual responses. This study could be a useful guide for practitioners on how to use wearable personal cooling systems to improve individual thermal comfort in warm indoor environments.

## 2. Methodology

### 2.1. Participants

Assuming an effect size of 0.65, a significance level of $\alpha = 0.05$, and a power of 0.8, 11 participants could provide enough power to detect a statistical difference of comparable magnitude (G*Power Version 3.1.9.6, Heinrich-Heine-Universität Düsseldorf, Düsseldorf, Germany). As a result, 16 young college students (8 males and 8 females) participated in this project. The physical characteristics of 16 participants are shown in Table 1. All of the participants were physically healthy and had no history of heat illnesses, pulmonary, or cardiovascular diseases. They were advised not to drink tea, coffee, alcohol and not to engage in any strenuous activity for at least a day before each trial. Participants were fully briefed on the purpose and details of this study prior to their participation. Following that, a written informed consent was obtained. Participants are free to leave the study at any time without penalty. After completing all trials, they were given an honorarium.

**Table 1.** Physical characteristics of participants.

| Gender | Age (yr) | Height (m) | Weight (kg) | Body Mass Index (kg/m$^2$) | Body Surface Area (m$^2$) |
|---|---|---|---|---|---|
| Males | $23.8 \pm 1.7$ | $1.76 \pm 0.04$ | $67.13 \pm 8.29$ | $21.70 \pm 1.85$ | $1.82 \pm 0.13$ |
| Females | $22.6 \pm 2.0$ | $1.64 \pm 0.05$ | $55.13 \pm 3.87$ | $20.62 \pm 1.71$ | $1.59 \pm 0.07$ |
| Overall | $23.1 \pm 2.3$ | $1.70 \pm 0.08$ | $66.13 \pm 8.80$ | $21.16 \pm 1.81$ | $1.70 \pm 0.15$ |

Note: data are presented as mean $\pm$ SD (standard deviation).

### 2.2. Face and Neck Cooling Fans

In order to investigate the actual performance of energy-efficient wearable cooling fans on the improvement of occupant thermal comfort in warm indoor conditions, two commercially available wearable U-shaped cooling fans were chosen: a face cooling fan and a neck cooling fan (see Figure 1). The face cooling fan (Gusgu, Shenzhen Gushang Digital Co., Ltd., Shenzhen, China) generates airflow by using two brushless 360° rotating direct current axial fans and can be worn around the neck. The two axial fans have a diameter of 7.5 cm and they can be controlled at three different speeds. This wearable face cooling fan has a built-in rechargeable lithium battery with a capacity of 2000 mA·h (7.4 W·h, voltage: 3.7 V) that can be recharged via USB. The cooling duration varies depending on the speed level (air speed ranged from 2.20–4.00 m/s at levels 1–3 [measured with an anemometer at a distance of 10 cm]). The total aerodynamic noise produced by the two axial fans in the facial cooling fan when operating at maximum fanning speed was determined to be 70.1 dB. Hence, the quality of the facial cooling fan is 'Good' based on the machine vibration nomogram. The face cooling fan's total power and weight are 4.0 W and 220 g, respectively. Two 5-cm (diameter) brushless direct current centrifugal fans were used for the wearable neck cooling fan (Gusgu WT-F41, Shenzhen Gushang Digital Co., Ltd., Shenzhen, China). The air produced by the two centrifugal fans is expelled

through 76 tiny vents located along the U-shaped air ducts. This type of neck cooling fan is powered by a rechargeable lithium battery and the cooling duration with a capacity of 2400 mA·h (8.9 W·h, voltage: 3.7 V) and has a cooling duration of 3–16 h. This neck cooling fan, similar to the wearable face cooling fan, has three air speed settings (averaged air speed at the outlets was 1.15–3.25 m/s at speed levels 1–3). The total noise produced by the two centrifugal fans in the neck cooling fan when operating at maximum fanning speed was determined to be 62.8 dB. Hence, the quality of the neck cooling fan is 'Good' based on the machine vibration nomogram. The neck cooling fan's total power and weight are 3.7 W and 260 g, respectively.

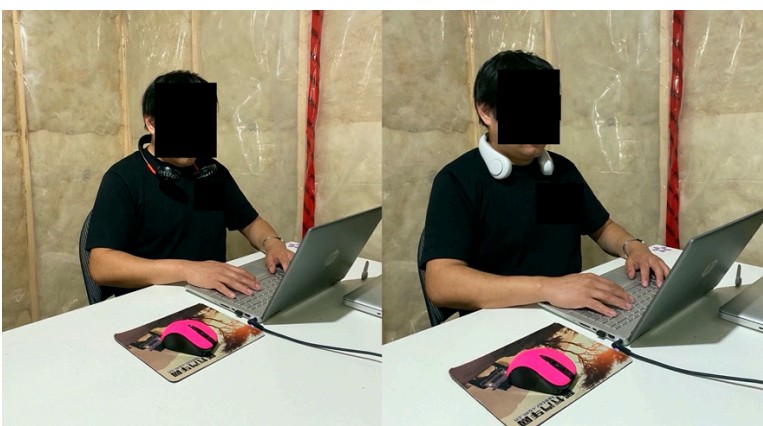

**Figure 1.** Facial cooling fan (**Left image**) and neck cooling fan (**Right image**).

### 2.3. Test Protocol and Procedure

Each participant completed 12 trials [2 temperature × 3 cooling options × 2 cooling modes] at two levels of air temperature (i.e., 30 and 32 °C), with three cooling options (i.e., CON [no cooling], FC [face cooling using the face cooling fan] and NC [neck cooling using the neck cooling fan]), and two cooling control modes (i.e., fixed power at the speed level 2 [fixed], and freely control the fan speed [free-control, the fanning speed of the facial and neck cooling fans was initially set to 0 m/s). The selection of the speed level 2 in the fixed-power mode was chosen based on participant feedback from a pilot trial regarding the most frequently used fanning speed during practice. As a result, there are 192 test scenarios in total. Every trial was randomized, counter-balanced and carried out at the same time of day.

Participants rested in armchairs for 20–30 min after arriving at the laboratory. They were then instrumented with skin temperature sensors. Local skin temperatures at the forehead, face and the neck were measured using wireless skin temperature loggers (iButton DS1922L, Maxim Integrated, San Jose, CA, USA; resolution: 0.0625, accuracy: ±0.5 °C). Participants were dressed in underwear (briefs, panties, bra [for females]), long trousers, a short sleeve t-shirt (100% polyester), socks, a pair of shoes (estimated total clothing thermal insulation is 0.57 clo). They then entered the climatic chamber (dimension: 3800 × 3800 × 2600 cm³) and were seated around a table. Participants could choose to read books or work on computers throughout the entire trials (estimated metabolic rate was 1.0 met).

Throughout the trials, occupants' perceptual responses including overall and local-body thermal sensation votes (TSVs), overall thermal comfort votes (TCVs), dry eyes and lips were surveyed at 10-min intervals (detail of perceptual rating scales is addressed in Section 2.5). Each exposure trial lasts 50 min in total. The air temperature, relative humidity (RH), air speed and the carbon dioxide concentration were measured every 1 min inside the chamber. Local skin temperatures at the forehead, face and the neck were collected every 1 min as well. Table 2 illustrates the equipment used in the chamber to record environmental parameters.

**Table 2.** Details of measurement equipment used in this study.

| Parameters | Instrument (Type and Manufacturer) | Accuracy |
|---|---|---|
| Air temperature | HOBO U12–012 (Onset Corp., Bourne, MA, USA) | $\pm 0.35\,^\circ$C |
| Relative humidity | HOBO U12–012 (Onset Corp., Bourne, MA, USA) | $\pm 2.5\%$ |
| Air speed | Swema 03 anemometer (Swema AB, Farsta, Sweden) | $\pm 0.05$ m/s |
| $CO_2$ concentration | RTR-576 (T&D Corporation, Nagano, Japan) | $\pm 50$ ppm |

Note: ppm, parts per million.

*2.4. Test Conditions*

Two indoor air temperatures were chosen for this study, i.e., 30 and 32 °C. The operative temperature inside the chamber was assumed to be the same as the air temperature because the wall temperature was kept at the same temperature as the ambient air. The indoor relative humidity was maintained at $50 \pm 5\%$ and the air speed was 0.1 m/s. The partial water vapor pressure in the chamber was 2.16 and 2.42 kPa at 30 and 32 °C temperatures, respectively. According to the CBE thermal comfort tool [19], the PMV (predicted mean vote) was +1.55 and +2.29 at air temperatures of 30 and 32 °C, respectively.

*2.5. Perceptual Response Questionnaire*

E-questionnaire was used to collect overall and local-body perceptual responses from occupants. Overall perceptual responses included the ASHRAE 7-point thermal sensation vote (TSV) [20], thermal comfort vote (TCV), and ratings of dry eyes and dry lips [10]. The TSV scale ranged from 'Cold' (−3), to 'Cool' (−2), to 'Slightly cool' (−1), to 'Neutral' (0), to 'Slightly warm' (+1), to 'Warm' (+2), to 'Hot' (+3). TCV scale went from 'Very uncomfortable' (−3), to 'Uncomfortable' (−2), to 'Slightly uncomfortable' (−1), to 'Neutral' (0), to 'Slightly comfortable' (+1), to 'Comfortable' (+2), and to 'Very comfortable' (+3). Ratings of dry eyes and dry lips ranged from 'Dry' (−2), to 'Slightly dry' −1), to 'Neutral' (0), to 'Slightly wet' (+1), to 'Wet' (+2). All rating scales are continuous except for the rating scales for dry eyes and dry lips (which are discrete). Throughout the trials, the questionnaire appeared automatically on occupants' computer screens every 10 min, and data were saved in the computer. All survey questions were completed in about 60 s by participants.

*2.6. Statistical Analysis*

The last 20 min of each trial's steady-state data were analyzed and reported. The Shapiro-Wilk test was used to determine normality of the data, which is reported as mean $\pm$ SD (standard deviation). Violations of Mauchly's test of sphericity were adjusted using Greenhouse-Geisser adjustments. A three-way repeated-measures ANOVA was performed to see if the independent variables (i.e., cooling conditions [CON, FC and NC], air temperatures [30 and 32 °C] and cooling modes [fixed and free-control]) had any effect on dependent variables, which included local skin temperatures at the forehead, face and the neck, as well as overall and local perceptual responses. If a significant difference was detected, Paired Samples t-tests were used to determine which pairs of test scenarios differed. All statistical analyses were carried out using SPSS Statistics Version 26.0 (IBM, Chicago, IL, USA). The level of significance was set at $p < 0.05$.

**3. Results**

*3.1. Local Skin Temperatures at Forehead, Face and Neck*

Local mean skin temperatures at the forehead, face and neck in the 12 testes scenarios are presented in Figure 2a–c, respectively. In both two studied air temperatures (i.e., 30 and 32 °C), the use of wearable face and neck cooling fans significantly reduced local skin temperatures at the forehead, face and the neck ($p < 0.001$). The forehead temperature decreased by 0.3–1.0 °C in FC and NC when compared to CON. At 30 °C, free-control of the fans raised local skin temperatures at the forehead, with the mean forehead temperature in FC30(free-control) and NC30(free-control) being 34.8 and 35.4 °C, respectively. In contrast,

no significant difference was noted in the mean forehead temperature between NC32(fixed) and NC32(free-control).

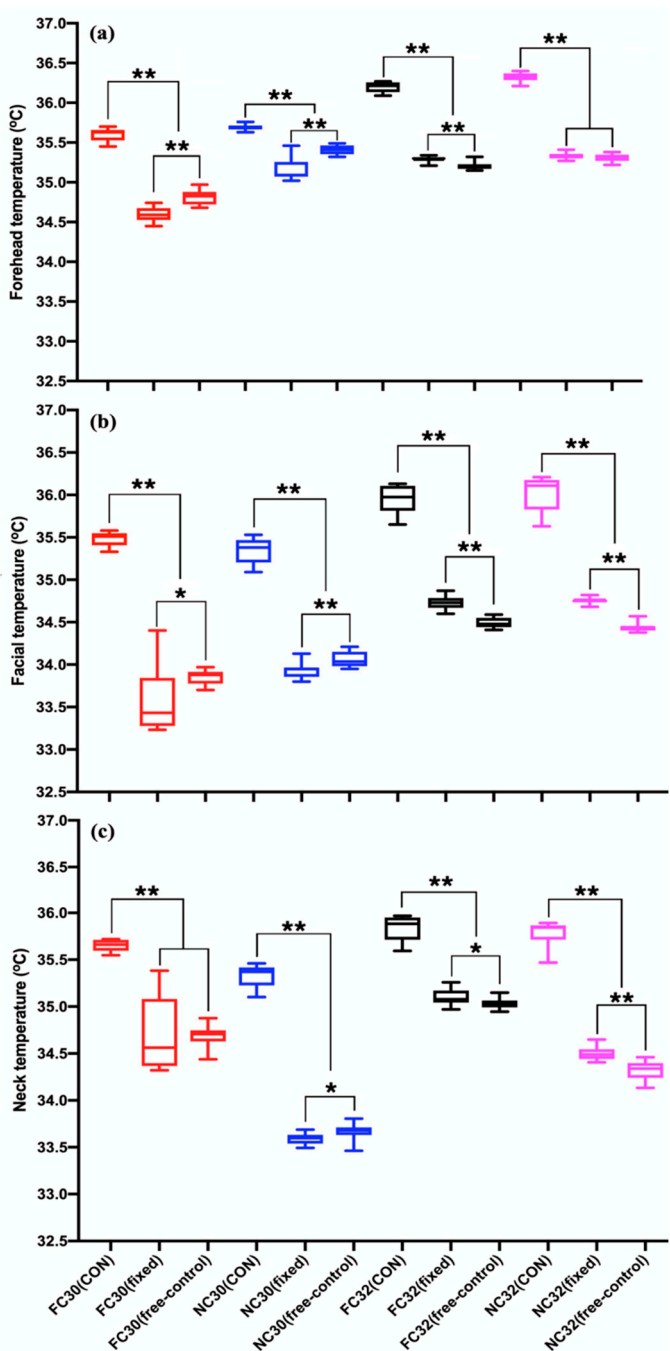

**Figure 2.** Local mean skin temperatures at the (**a**) forehead, (**b**) face and (**c**) neck of 12 studied test scenarios. *, $p < 0.05$; **, $p < 0.001$.

With regard to the mean face temperature, it was reduced by 1.4–1.9 °C at 30 °C in FC and NC when compared to no cooling. Similarly, the mean face temperature dropped by 1.2–1.6 °C at 32 °C temperature when the two wearable cooling fans were applied. The mean face skin temperature was 34.7, 34.5, 34.8 and 34.4 °C in FC32(fixed), FC32(free-control), NC32(fixed) and NC32(free-control), respectively. Furthermore, it was discovered that local face temperature was significantly lower in face cooling fan scenarios than the neck cooling fan scenarios.

The use of face cooling fan could only reduce the neck temperature by 0.7–1.0 °C, whereas the use of neck cooling fan could reduce the local neck skin temperature by 1.3–2.1 °C. Significant differences in mean neck temperature were observed between the fixed-power mode and the free-control mode in NC30 ($p < 0.001$), FC32 ($p < 0.05$) and NC32 ($p < 0.001$). The mean neck skin temperature was 34.7 and 34.7 °C in FC30(fixed) and FC30(free-control), respectively. It was 33.6 and 33.7 °C in NC30(fixed) and NC30(free-control), respectively. At 32 °C, the mean neck skin temperature was 35.1, 35.0, 34.5 and 34.3 °C in FC32(fixed), FC32(free-control), NC32(fixed) and NC32(free-control), respectively.

### 3.2. Overall Thermal Sensation Vote (TSV)

Overall thermal sensation votes (TSVs) of the 12 tested scenarios are shown in Figure 3. The observed overall TSVs were +1.73 (close to 'Warm'), +1.66 (close to 'Warm'), +2.44 (in between 'Warm' and 'Hot') and +2.40 (in between 'Warm' and 'Hot') in FC30(CON), NC30(CON), FC32(CON), NC32(CON), respectively. The use of two energy-efficient cooling fans significantly improved overall TSVs in all test scenarios ($p < 0.001$). Overall TSVs decreased to +0.59 to +0.64 (in between 'Neutral' and 'Slightly Warm') when face and neck cooling fans were used at 30 °C. When face and neck cooling fans were used at 32 °C, it dropped to +0.74 to +1.10 (close to 'Slightly Warm'). Furthermore, in all test scenarios, the free-control mode had no significant effect on the overall TSVs when compared to the fix-power mode ($p > 0.05$), with the exception of the NC32 scenarios ($p < 0.05$).

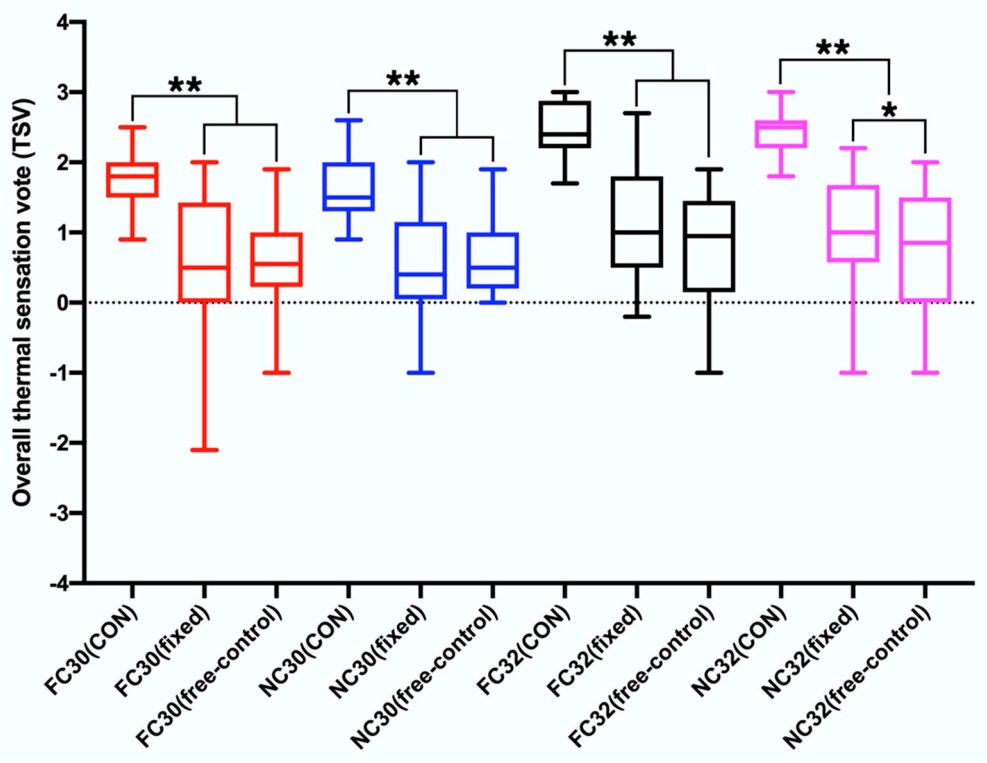

**Figure 3.** Overall thermal sensation votes (TSVs). *, $p < 0.05$; **, $p < 0.001$.

### 3.3. Overall Thermal Comfort Vote (TCV)

Overall thermal comfort votes (TCVs) of the 12 tested scenarios are displayed in Figure 4. Overall TCVs were −0.85 and −0.65 (close to 'Slightly Uncomfortable') when no cooling was used at 30 °C. At 32 °C in CON, overall TCVs were −1.12 and −1.13 (close to 'Slightly uncomfortable'). The use of two wearable cooling fans significantly improved overall TCVs when compared to no cooling ($p < 0.05$ or $p < 0.001$). Overall TCVs were improved by 0.30–0.96 scale unit when the two wearable cooling fans were used at both air temperatures. It is interesting to note that the free-control mode could significantly improve

overall TCVs when compared to the fixed-power mode at 30 °C ($p < 0.05$). Overall TCVs in FC30(free-control) and NC30(free-control) were +0.11 and −0.07 (close to 'Neutral'), respectively. At 32 °C, no significant differences in overall TSVs were found between the fixed-power and free-control modes with either cooling fan ($p > 0.05$).

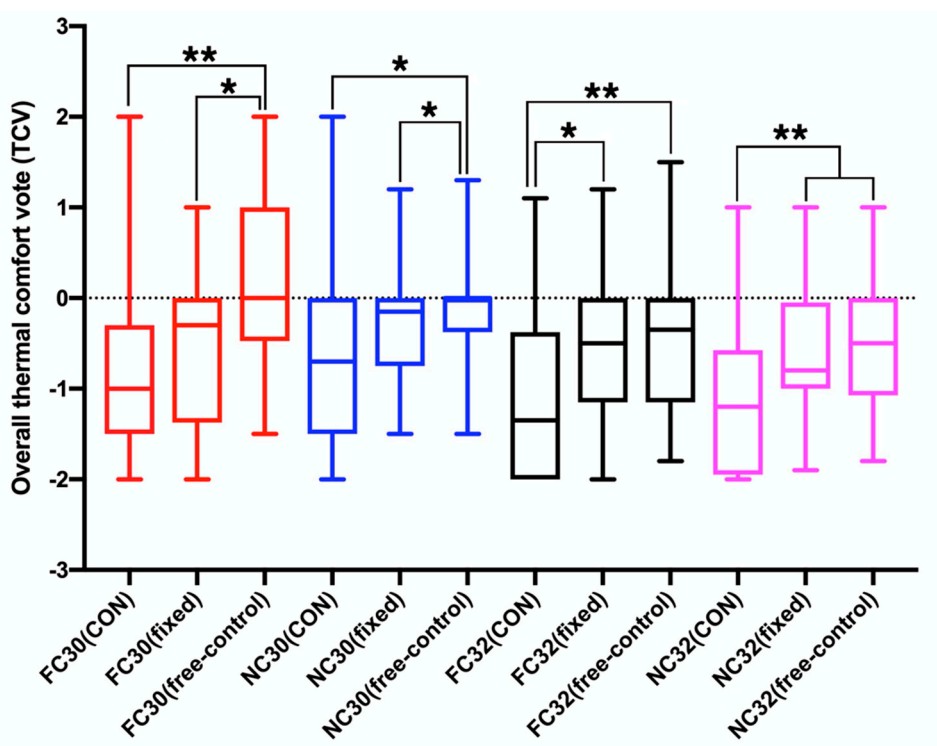

**Figure 4.** Overall thermal comfort votes (TCVs). *, $p < 0.05$; **, $p < 0.001$.

### 3.4. Local Thermal Sensation at Face and Neck

Local thermal sensation votes (TSVs) at the face and the neck areas are illustrated in Figure 5a,b, respectively. Compared to no cooling, the two wearable cooling fans significantly improved local thermal sensation at the face and neck ($p < 0.001$). At 30 °C, cooling fans reduced local TSVs at the face area by 0.78–1.20 scale unit, whereas at 32 °C, cooling fan reduced it by 0.98–1.21 scale unit. All local TSVs at the face area were kept below +0.61 (in between 'Neutral' and 'Slightly warm') during the cooling period at both 30 and 32 °C. Furthermore, the face cooling fan outperformed the neck cooling fan on local TSVs at the face. In NC30 scenarios, the free-control mode demonstrated significantly higher local TSVs at the face than the fix-power mode ($p < 0.05$).

When no cooling was applied, mean local TSVs at the neck ranged between +1.19 and +1.62 (close to 'Slightly warm') (see Figure 5b). Both cooling fans reduced local TSVs at the neck area by more than 0.82 scale unit. Local TSVs at the neck were +0.29, +0.43, +0.24 and +0.36 (all values were close to 'Neutral') in FC30(fixed), FC30(free-control), NC30(fixed) and NC30(free-control), respectively. Similarly, local TSVs) at the neck in FC32(fixed), FC32(free-control), NC32(fixed) and NC32(free-control were +0.55, +0.44, +0.52 and +0.37 (in between 'Neutral' and 'Slightly warm'), respectively. As a result, the neck cooling fan may induce greater improvement on local TSVs at the neck than the face cooling fan. At 32 °C only, the free-control mode had significantly lower TSVs at the neck than the fix-power mode ($p < 0.05$).

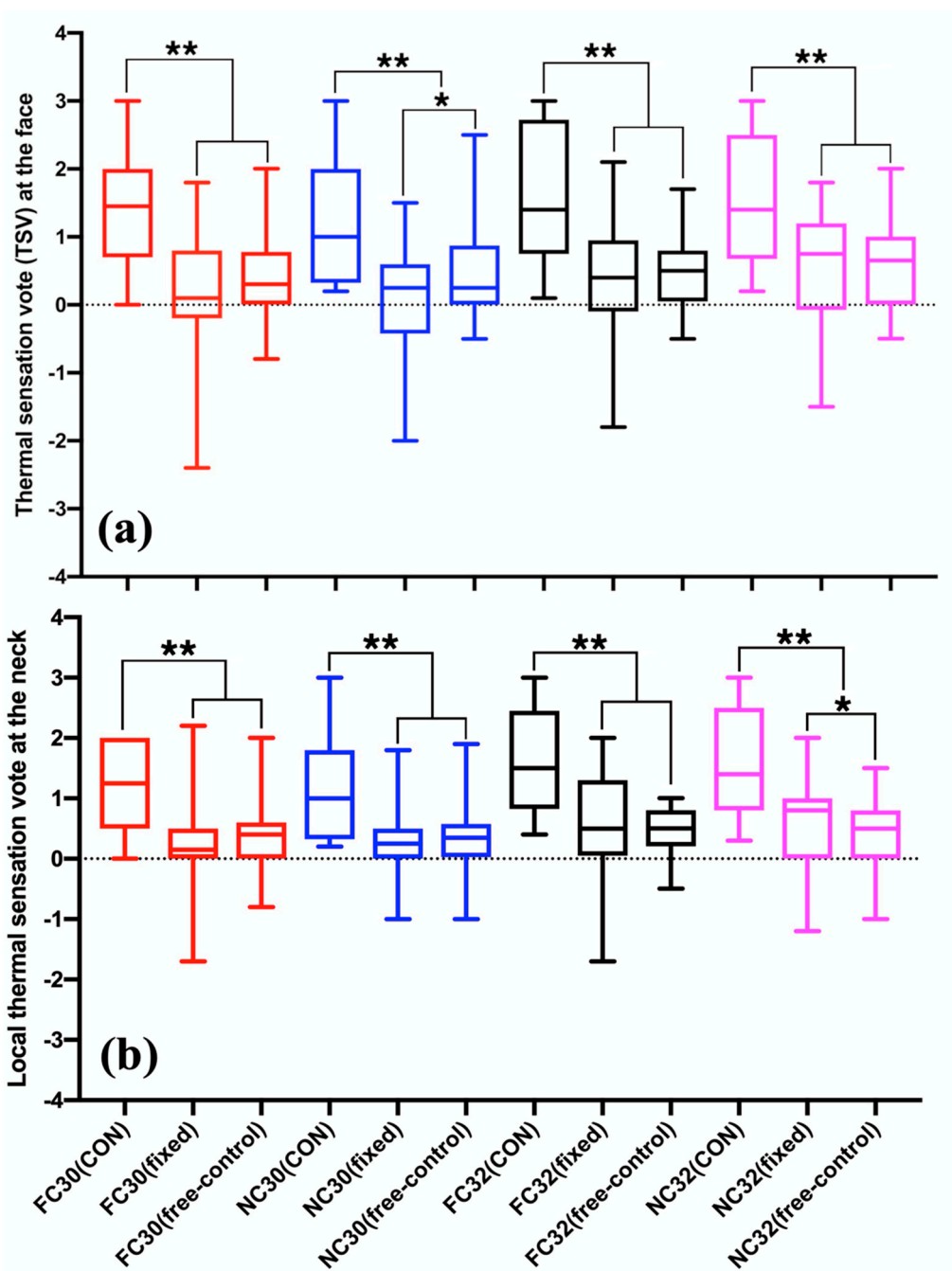

**Figure 5.** Local thermal sensation votes at (**a**) the face area and (**b**) the neck. *, $p < 0.05$; **, $p < 0.001$.

### 3.5. Dry Eye and Dry Lip Syndromes

Ratings of dry eyes and dry lips in the 12 tested scenarios are demonstrated in Figures 6 and 7, respectively. Figure 6 showed that using both cooling fans increased the percentage of participant who reported having dry eye syndromes (ratings of −2 and −1). For example, when no cooling was used, only 12.5–25% of participants reported dry eye syndromes, whereas when the two wearable cooling fans were used, the percentage of participants reporting dry eye syndromes of −2 and −1 increased to 50–68.8%. On the other hand, the face cooling resulted in up to 15.6% more syndromes of dry eyes than the neck cooling fan. The free-control mode improved dry eye symptoms by 18.8% when compared to the fix-power mode with the face cooling fan at 30 °C. In comparison to the fixed-power mode, the free-control mode worsened the dry eye syndromes by 6.25% with the face cooling fan at 32 °C.

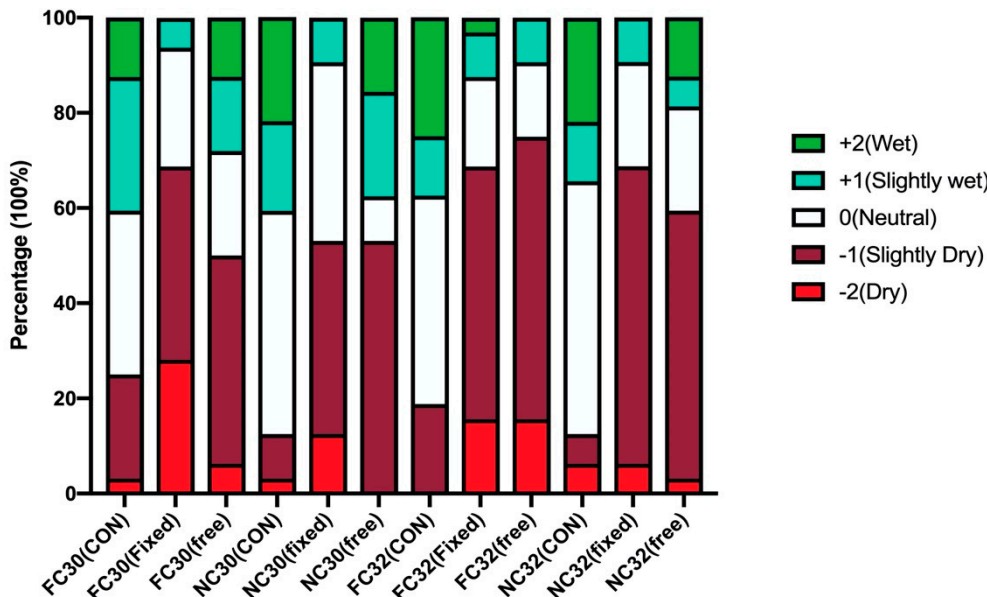

**Figure 6.** Ratings of dry eye symptoms in the 12 tested scenarios.

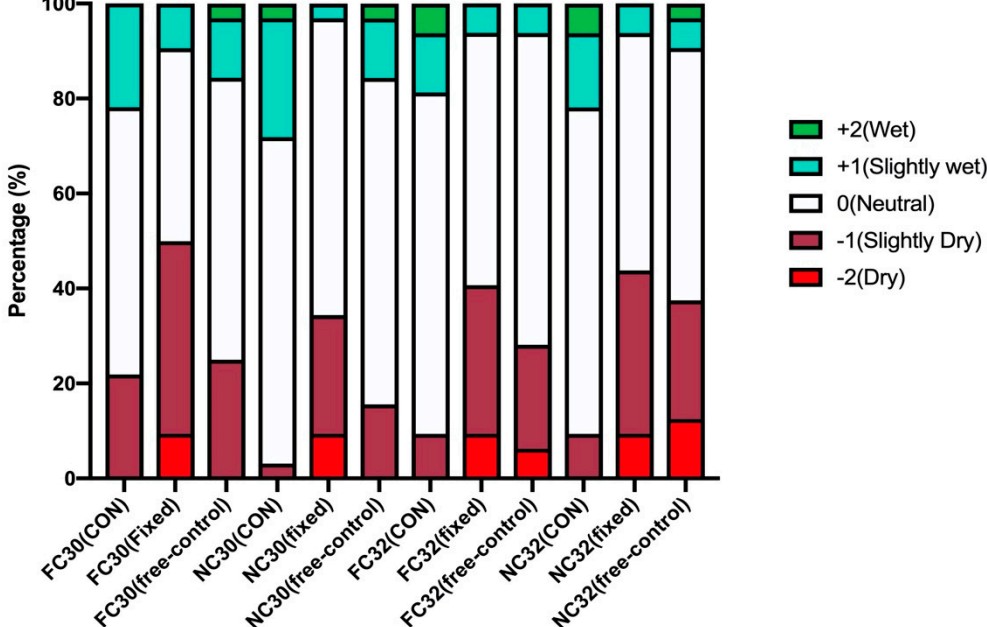

**Figure 7.** Ratings of dry lip symptoms in the 12 tested scenarios.

In terms of ratings of dry lip symptoms, the use of two wearable cooling fans significantly worsened the problem when compared to no cooling (i.e., only <21.9% of participants reported dry lip symptoms [see Figure 7]). The percentage of dry lips raised from 9.4–21.9% to 15.6–50.0% when cooling fans were used. At 30 °C, the neck cooling fan reduced dry lip symptoms by 9.4–15.6% more than face cooling fan. In all cooling fan cases, the free-control cooling mode greatly improved the dry lips issue by 6.3–18.8% when compared to the fixed-power cooling mode.

## 4. Discussion

The surface area of the human face and neck accounts for only 3.5% and 2% of an adult's total body surface area, respectively [21]. Cooling the face and neck has been widely used to improve athletic performance in the heat [22–25]. Neck cooling during exercise, in

particular, has been shown to improve exercise performance in the heat [23,26,27]. Despite the fact that the activity intensity was quite high during exercise and sports, the neck and face cooling functioned well. The activity level of occupants in indoor environments was much lower than that of athletes. As a result, cooling the face and neck should be even more effective at improving thermal comfort for indoor occupants than it is for athletes.

To the best of our knowledge, this is the first study to look into the effect of wearable face and neck cooling on thermal comfort of indoor occupants. Previous research on electric fans [13,28–32] has consistently shown that electric fans can improve occupants' thermal comfort in a variety of warm indoor conditions. Nevertheless, such electric fans exposed the occupants to either whole-body convective cooling or upper-body cooling (including face and neck cooling). However, the effectiveness of face and neck cooling in improving occupant thermal comfort in indoor environments has remained unknown. The findings of this study showed that wearable face and neck cooling fans could significantly reduce local skin temperatures at the forehead, face and neck region by up to 2.1 °C. It was also discovered that face cooling fan could result in higher temperature reduction at the forehead and the face when compared to the neck cooling fan. In contrast, the neck cooling fan may cause a greater reduction in skin temperature at the neck than the face cooling fan (see Figure 1). Furthermore, local thermal sensation votes at the face improved by up to 1.21 scale units at the two studied air temperatures (i.e., 30 and 32 °C). Similarly, the wearable face and neck cooling fans reduced the local thermal sensation at the neck by over 0.82 scale unit. Zhang and Zhao [33] investigated the effect of face cooling (supplied by a personalized ventilation system) on human responses and discovered that while face cooling was provided, the acceptable room temperature range could be increased from 26 to 30.5 °C. In this study, it appeared that the acceptable room temperature could be raised to 32.0 °C by using wearable face and neck cooling fans. Therefore, this could result in an average savings of 45.7% when compared to the baseline HVAC setpoint of 24.5 °C [34].

Energy-efficient wearable face and neck cooling fans (power consumption $\leq 4$ W) significantly improved not only local thermal sensation at the face and the neck, but also significantly improved the overall thermal sensation as well as overall thermal comfort. Overall TSVs ranged from +1.66 to +1.73 at 30 °C air temperature and +2.40 to +2.44 at 32 °C. The PMV predicted by the CBE thermal comfort tool was +1.55 and +2.29 at those two indoor temperatures (30 and 32 °C), respectively. Our observed TSVs were clearly in good agreement with PMVs predicted by the CBE thermal comfort tool [19]. When the wearable face and neck cooling fans were used, overall TSVs decreased by 1.03-1.14 scale unit at 30 °C and by 1.34 to 1.66 scale units at 32 °C. The observed overall TSVs were close to +0.5 ('Slightly warm') when face and neck cooling fans were used at 30 °C, indicating that approximately 90% of participants were satisfied with the thermal environment [35]. At 32 °C, the observed overall TSVs were close to +1.0, indicating that approximately 26% occupants were dissatisfied with the thermal condition (with cooling, PPD [Predicted Percentage of Dissatisfied] = 88%). Thus, the use of face and neck cooling fans could result in 74% of occupants being satisfied with the thermal environment. Nevertheless, this figure is slightly lower than the ASHRAE 55 standard of 80% occupant satisfaction rate [20]. This could be due to the dry eye and lip symptoms caused by the use of face and neck cooling fans (see Figures 5 and 6). On the other hand, overall TCVs at 32 °C were all above −0.72 (see Figure 4), indicating that the 32 °C temperature was still acceptable when using such highly energy efficient wearable face and neck cooling fans.

The effect of cooling control mode on thermal comfort has also been investigated in this work. At 30 °C, the fixed-power control mode caused overcooling in the face and neck regions, as evidenced by lower overall and local TSVs as well as the lower TCVs when compared to the free control mode (see Figures 2–5). In contrast, at 32 °C, the fixed-power at the speed level 2 (corresponding wind speed: 2.18 m/s) was unable to provide adequate cooling to the occupants. As a result, the fixed-power mode worsened overall and local TSVs as well as overall TCVs when compared to the free-control mode. The findings above confirmed that personal control (free-control mode) played an important role in improving

individual thermal comfort. This is consistent with previous research demonstrating that individual control of personal comfort devices can improve an individual's satisfaction with indoor conditions as well as energy efficiency [36–42].

## 5. Limitations and Future Perspective

Some limitations of this study should be acknowledged. First, only young college students were recruited, limiting our findings to populations of various ages and vulnerable groups. Second, local thermal comfort at the face and the neck was not investigated, despite the fact that such details could provide useful information for investigating its impact on overall thermal comfort. Next, only natural air cooling was studied, with no consideration given to other wearable cooling options such as liquid cooling and evaporative cooling.

Future research should be conducted to assess the efficacy of such energy efficient face and neck cooling fans on the elderly in a variety of temperature conditions. The elderly adults have low thermal sensitivity when compared to young adults and they are not sensitive to temperature changes [43]. Furthermore, the elderly has impaired sweating capacity. A previous study [44] has shown that the fan use had no or negative benefits on the elderly aged 60 to 80 in extreme heat, the use of wearable fans such as those used in our study on thermal comfort improvement of the elderly remains unknown. Furthermore, different types of wearable cooling devices should be investigated in order to identify the best performance wearable cooling devices for improving occupant thermal comfort in higher temperature indoor environments where HVAC systems are not available. Lastly, the impact of prolonged use of wearable fans on dry eye and dry lip syndromes should be extensively investigated in future research. It is expected that the fan use reduces humidity in the near-head area and/or hasten tear evaporation, resulting in dry eye (lip) symptoms. In addition, because of personal risk factors such as medication status and the use of personal care products, the elderly is more vulnerable to the development of dry eye syndromes [45]. Hence, it is possible that using facial and neck cooling fans may make dry eyes and dry lips more common in the elderly. Nonetheless, the aforementioned hypotheses require further investigation.

## 6. Conclusions

Two highly energy efficient wearable face and neck cooling fans were used to improve occupant thermal comfort while performing office work in two warm indoor environments. Occupants' physiological and perceptual responses while using these two types of wearable cooling fans were studied and compared. Results showed that both wearable cooling fans could largely reduce local skin temperatures at the forehead, face and neck regions by up to 2.1 °C. Local thermal sensation votes at the face and the neck were reduced by 0.82–1.21 scale units. Overall TSVs decreased by 1.03–1.14 scale units at 30 °C and by 1.34–1.66 scale units at 32 °C. Both cooling fans could raise the acceptable HVAC temperature setpoint to 32.0 °C, resulting in a 45.7% energy saving over the baseline HVAC setpoint of 24.5 °C. Furthermore, occupants are advised to use the free-control cooling mode when using those two types of wearable cooling fans to improve indoor thermal comfort. Despite some issues with dry eye and dry lip symptoms, it is ultimately concluded that the two types of wearable cooling fans chosen could significantly improve indoor thermal comfort and save HVAC cooling energy.

**Author Contributions:** Conceptualization, F.W.; methodology, F.W. and B.Y.; software, F.W.; validation, T.-H.L. and K.L.; formal analysis, B.Y. and F.W.; investigation, P.Y.; resources, B.Y.; data curation, P.Y.; writing—Original draft preparation, B.Y. and F.W.; writing—Review and editing, T.-H.L.; K.L. and F.W.; supervision, B.Y. and F.W.; project administration, F.W.; funding acquisition, B.Y. All authors have read and agreed to the published version of the manuscript.

**Funding:** This research received no external funding.

**Institutional Review Board Statement:** Not applicable.

**Informed Consent Statement:** Informed consent was obtained from all subjects involved in the study.

**Data Availability Statement:** The data presented in this study are available upon request from the corresponding author.

**Conflicts of Interest:** The authors declare that they have no known competing financial interest or personal relationships that could have appeared to influence the work reported in this paper.

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
