# Peer review of "On the Use of Wearable Face and Neck Cooling Fans to Improve Occupant Thermal Comfort in Warm Indoor Environments"

_energies, doi:10.3390/en14238077_

Round 1

Reviewer 1 Report

The study investigates the impact of using energy-efficient wearable face and neck cooling fans on occupant thermal comfort in warm indoor environments.

Overall, the article is very well written, and it is divided into clear and coherent sections. Figures and tables are clearly presented and cited in the text. 

The aim of the study is very interesting and worth investigating, given the importance of occupant comfort when dealing with energy-efficient measures, and the energy savings that could be achieved based on the presented results.

I only have two comments to be addressed by the authors:

  1. given that the participants are males and females, it was worth presenting some specific analysis by gender. So, it is either another limitation of the study, or the authors can add some discussion regarding it.
  2. Given the Metabolic rate of the participants and the clothing levels, it would be worth adding some recommendations on the applicability of the investigated systems by building type (also the obtained thermal comfort dissatisfaction can help by comparing to the categories proposed by the standards).

Author Response

The study investigates the impact of using energy-efficient wearable face and neck cooling fans on occupant thermal comfort in warm indoor environments. Overall, the article is very well written, and it is divided into clear and coherent sections. Figures and tables are clearly presented and cited in the text. The aim of the study is very interesting and worth investigating, given the importance of occupant comfort when dealing with energy-efficient measures, and the energy savings that could be achieved based on the presented results.

[REPLY] Thank you for your feedback.

I only have two comments to be addressed by the authors:

1.given that the participants are males and females, it was worth presenting some specific analysis by gender. So, it is either another limitation of the study, or the authors can add some discussion regarding it.

[REPLY] That is an excellent suggestion. Given the small number of male and female participants and the fact that local thermal comfort was not studied in this work, we decided to include a section to discuss the future perspective and practical applications (see the section 6 Future perspective and practical applications, lines 408-461), which includes the effect of facial and neck cooling fans on different genders. See below the added new text:

More research is needed to fully understand the gender differences in cooling demand for facial and neck cooling using wearable fans. Females are more sensitive to temperature change than males, according to documented research [43,44]. In thermal comfort responses to slightly warm environments, there were few gender differences [43]. Female occupants are less satisfied with room temperatures than male occupants and they prefer higher room temperatures than males[45]. Further, female occupants are more likely than males to experience uncomfortably hot conditions, but females adjust thermostats in the home less frequently than males. In light of the foregoing evidence, the use of wearable facial and neck cooling fans may help to alleviate the different thermal demands caused by gender differences. As a result, improved comfort for both male and female office workers through the use of facial and neck cooling fans may increase productivity in some of their tasks.

     Given that females have different thermal demands in room temperatures that deviate from thermoneutral, it is expected that if facial and neck cooling fans are operated in fixed power mode, there will be a difference in thermal comfort between the two genders. Nonetheless, more research is needed to confirm this. Local thermal comfort differences when using facial and neck cooling fans between genders, on the other hand, has not yet been thoroughly investigated. Extensive research on local thermal comfort of both genders could reveal the true thermal demands on the use of facial and neck cooling fans for both genders. Thus, more research is required to determine the differences in local and overall thermal comfort between the genders when using facial and neck cooling fans in both fixed power and free control modes.

  1. Given the Metabolic rate of the participants and the clothing levels, it would be worth adding some recommendations on the applicability of the investigated systems by building type (also the obtained thermal comfort dissatisfaction can help by comparing to the categories proposed by the standards).

[REPLY] Thank you for your helpful advice. We have added a paragraph about practical applications, which can be found below the added text:

In terms of practical applications, factors such as adjustability, adjustment response time, cost, user willingness to use or purchase the facial and neck cooling fans, and the potential application of facial and neck cooling fans to each building type should all be considered. According to the findings of this work, facial and neck cooling fans can be easily applied to commercial offices, hotels, residential houses and dwellings, as well as school buildings with limited personal occupation space. Wearable facial and neck cooling fans is also useful and practical in other types of buildings such as shopping malls, hospitals as well as industrial warehouses. Regarding the target populations, wearable facial and neck cooling fans can be used on both inactive occupational groups such as office workers, students, residents, shoppers, and so on. However, the true performance of using facial and neck cooling fans on people with higher metabolic rates should be thoroughly investigated in a future study.

Reviewer 2 Report

This paper presents the results of measurements of local skin temperatures and perceptual responses while using the two wearable cooling fans. The paper is clear and well written. I think it is an interesting article to publish in Energies journal.

Here are some comments that I hope will help to improve this paper:

1. The title is a bit misleading. ‘Use of highly energy efficient wearable face and neck cooling fans...’. This part of the title suggests that the energy consumption was analyzed, which, in fact, was not. I suggest editing the title.

2. What noise is generated by the fans and can this cause discomfort? Comment on it in the text.

3. Has the weekly / monthly / annual energy consumption of these devices been estimated?

4. In Section 3.1 the changes in face temperature are given. Values are below the measurement accuracy. Can they be considered correct in such situation?

Author Response

See attached file, thanks.

Reviewer 3 Report

Regarding the equipment: a detailed photo of the equipment would be appreciated. Does the equipment produce any noise or vibration that is annoying?

Test protocol & procedure

Line 157: x 3 cooling x “2” cooling

Explain which are the two cooling modes in non-cooling CON mode

In free control, the initial fan speed must be indicated.

Perceptual response questionnaire

Are the ASHRAE 7-point thermal sensation vote (TSV): Very uncomfortable '(-3), to' Uncomfortable '(-2), to' Slightly uncomfortable '(-1), to' Neutral '(0), to' Slightly comfortable '(+1 ), to 'Comfortable' (+2), and to 'Very comfortable'?

 Results

Nothing is comented about whether in free control users are varying the fan speed, or in what direction they are doing it (increasing or decreasing), or if the change in speed increases their feeling of comfort.

Figs 3, 4, 5 and 6

How do you explain the different results between FC30 (CON) and NC30 (CON)?

How do you explain the different results between FC32 (CON) and NC32 (CON)?

Fig 6:

The results for FC32 (CON) and NC32 (CON) are the same, as can be expected. However, how do you explain the different results between FC30 (CON) and NC30 (CON)?

Is there variation in the voting as the test proceeds (from the first vote at 10 minutes to the last vote at 50 minutes)?

How are the results expected to be with older people?

How are the results expected to be with test times greater than 50 minutes (especially with regard to dry eye and dry lip syndromes)?

Author Response

See attached file, thanks.

Round 2

Reviewer 3 Report

No further comments.